# Citizen Science for Scientific Literacy and the Attainment of Sustainable Development Goals in Formal Education

**Miguel Ángel Queiruga-Dios [1,*,†], Emilia López-Iñesta [2,†], María Diez-Ojeda [1,†],**
**María Consuelo Sáiz-Manzanares [3,†] and José Benito Vázquez Dorrío [4,†]**

[1]  Department of Specific Didactics, Universidad de Burgos, 09001 Burgos, Spain; mdojeda@ubu.es
[2]  Department Didactics of Mathematics, Universitat de València, 46022 València, Spain; emilia.lopez@uv.es
[3]  Department of Health Sciences, Universidad de Burgos, 09001 Burgos, Spain; mcsmanzanares@ubu.es
[4]  Department of Applied Physics, Universidade de Vigo, 36310 Vigo, Spain; bvazquez@uvigo.es
[*]  Correspondence: maqueiruga@ubu.es
[†]  These authors contributed equally to this work.

**Abstract:** Curricular integration of the formal teaching of citizen science can bring to the classroom aspects of scientific literacy that encourage the involvement of citizens. In particular, these include non-epistemic aspects related to the sociology of science (which are often not transferred to the classroom). Furthermore, this practice raises awareness among students, and encourages them to become participants in the attainment of the United Nations' Sustainable Development Goals (SDGs). This article describes a proposal for the integration of a citizen science project into the secondary education curriculum that can be reproduced in any educational center. Eighty-three secondary school pupils (14–15 years old) took part in this research at a city-center school in Northern Spain. A questionnaire based on validated studies was created and used to analyze the changes in attitudes of pupils towards science and technology and their improvement in scientific literacy in terms of scientific processes and scientific situations. The results indicate a significant improvement in the attitudes towards science and technology among the participating learners, as well as a better understanding of scientific processes and situations. Likewise, the results reflect how the implementation of the citizen science project contributes to the SDGs.

**Keywords:** citizen science; secondary education; scientific literacy; curricular integration; innovation; science; gender; SDG

## 1. Introduction

One of the common aims of education systems is to achieve scientific literacy among pupils [1,2]. Scientific literacy is understood to be a process that allows students to face relevant problems that require the recollection of the scientific knowledge they have learned [3], with the most important component being the knowledge and comprehension of the Nature of Science (NoS). This is because the NoS is what citizens use to evaluate matters of science and technology [4].

However, formal education commonly conveys scientific knowledge that is already formulated [3], which means that students do not develop all the dimensions of scientific literacy. These dimensions include—in addition to a conceptual understanding of some basic notions of science—the understanding of the processes and methods used and an understanding of the NoS and the influences that science, technology, and society [4,5] have on each other. According to Kemp [6], scientific literacy means that a person knows and understands conceptual ideas, is capable of using information in a procedural way, and has values and principles that align with an affective dimension.

Among those three dimensions (Figure 1), teachers must ask themselves which concepts foster development in school science, and which can be developed with the various tools available to them.

## Procedural Dimension
PD1.  Self-learning science
PD2.  Use science in everyday life
PD3.  Apply science for social purposes
PD4.  Decode scientific communications
PD5.  Encode scientific communications
PD6.  Think scientifically
PD7.  Reason and argue
PD8.  Judge the validity of claims
PD9.  Take decisions
PD10. Solve problems
PD11. Integrate knowledge
PD12. Engage in inquiry
PD13. Use some of the tools of science

## Affective Dimension
AD1.  Appreciation for science
AD2.  Interest in science
AD3.  Inclination to stay up to date
AD4.  Inclination to monitor and act on SRSP*
AD5.  Objective, open mind and skepticism
AD6.  Ethical values
AD7.  Self-confidence to use science
AD8.  Appreciation of the world

*Science-Related Social Problems

## Conceptual Dimension
CD1.  Science concepts
CD2.  The physical world
CD3.  Science vocabulary
CD4.  Broad principles of science
CD5.  Scientific inquiry
CD6.  Relations of science with mathematics
CD7.  Limitations of science and technology
CD8.  The attempt of scientific / technological knowledge
CD9.  Science is a social activity
CD10. Science and technology are human efforts
CD11. The history of science
CD12. Relations between science and society
CD13. Relationships science to technology
CD14. Relations between science, technology and society

**Figure 1.** Elements of scientific literacy grouped according to conceptual, procedural, and affective dimensions; adapted from [6]. Teachers must consider which concepts can be developed in school science.

The educational/teaching proposal described and analyzed here seeks to integrate citizen science projects into the school curriculum to foster scientific literacy in all its dimensions—particularly in those that are commonly relegated to the background or are not considered directly, such as "non-epistemic matters related to the spheres of the internal and external sociology of science in empirical studies" [4], and to create awareness among pupils and encourage them to become participants in achieving the Sustainable Development Goals (SDGs) [7]. Projects of this type are not only of relative importance when training pupils, but can also become catalyzers for their curiosity and creativity, while at the same time creating attitudes of social responsibility, collaboration, and citizen participation.

## 2. Theoretical Framework

Defining a concept like literacy is highly complex, as this concept is continually changing and depends upon its social, cultural, and political contexts. In [8–10], the functional level of literacy was defined as that which allows simple texts to be read. Above that is the advanced level, which also entails the complex processing of information. After that, a continuum of levels can be established

according to the complexity of the texts involved. Furthermore, the breadth of learning and knowledge means that the word "literacy" is used metaphorically for concepts such as scientific literacy, digital literacy, and technological literacy [8,10], many of which are interconnected, with literacy in one area aiding the development and improvement of that in others. Thus, through examples, we can discuss digital technologies and the potential they possess to enrich and develop overarching skills to facilitate the creation of learning environments that allow learning to take place in connection with the real world [11], while at the same time facilitating the inclusion of pupils with difficulties.

Literacy becomes manifest by means of the applied use of the abilities or competences that pupils acquire during their school life, and subsequently throughout their lives as integrated citizens. Formal education is responsible for the attainment of universal literacy. For that reason, it must be connected with the real and ever-changing world, and allow for the critical development of pupils by fostering their values, attitudes, and contexts for reflection. This will make it possible to create responsible citizens, rather than only people with qualifications. In the face of the changes taking place due to technological evolution, it is more appropriate to talk of new dimensions to literacy, rather than new literacies; the type of literacy needed in the 21st century is media-based, digital, and multi-modal [9].

Within this context of definitions, we can determine the meaning of scientific literacy, or the dimension of literacy linked to scientific–technological knowledge, which is a general concept with a variety of connotations [1,2,6,12,13] and variable meanings [14,15]. This term was coined by Paul Hurd at the end of the 1950s, to describe an understanding of science and its applications in society [2,14–16].

Starting with the basic concept of literacy as the capacity to read and write, scientific literacy can be defined as the capacity to read and write about science and technology [12]. Although this is different to other literacies, the capacity for comprehension, interpretation, and analysis are similar to those required to deal with other types of texts [17]. For example, scientific writing involves a series of specific lexicogrammatical resources that interact in a synergetic way to construct an interpretation of the natural world that is distinct from the one offered by everyday language [17,18]. Reading science texts involves decoding words to provide them with information and meaning, while paying attention not only to the substantive contents of the texts but also to the scientific meanings of statements and the roles of those statements in the reasoning that binds the elements comprising the substantive contents [17]. Reading such texts requires a metacognitive capacity for examining not only the sources of knowledge, their limits, and their certainty, but also for interpreting them and analyzing them critically; that is, it requires a mastery of literate thought in a general sense. Literacy has two related meanings: First, in its fundamental sense, it refers to the capacity to read and write, and the capacity to comprehend, interpret, and critically analyze scientific texts; second, in its derived sense, it refers to having knowledge of the significant contents of science and the connections that exist between the various concepts [17].

Therefore, scientific literacy can be measured by one's comprehension of the scientific approach, comprehension of basic scientific constructs, or comprehension of matters of scientific policy [12–14,19]. These types of comprehension are related to the three dimensions defined in [20,21]—the understanding of scientific concepts, processes, and situations [22]—and described specifically as the capacity to become engaged in issues related to science and scientific ideas in a reflective way, with a willingness to participate in debates on science and technology and the capacity to explain phenomena scientifically, evaluate and design scientific research, and interpret data and tests.

As stated in [23], "Scientific literacy also includes understanding the nature of science, scientific projects, and the role of science in society and one's own personal life", thereby involving the capacity to address and evaluate evidence-based scientific arguments and draw conclusions appropriately [19,23,24]. Furthermore, scientific literacy involves the capacity to identify the scientific problems underlying political decisions, and to express coherent viewpoints using scientific and technological arguments [23,25], given that a society's development, in terms of its citizenship, is closely linked to

its strength as a democracy [1,12,13,23]. Thus, scientific literacy is conceived by some authors as a property of collective activity, i.e., an indeterminate result of conversational activity [26].

In this interaction and dialogue between the scientific–technological world of scientists and their social context, an important role is played by citizen science, which, on the one hand, promotes the scientific literacy of citizens and, on the other, provides opportunities for scientists to convey to society how social concerns and demands influence their work [27], by showing that science and technology are important and accessible to all members of society. In this way, a new possibility is opened-up to construct knowledge through the participation of social agents in the processes of scientific production, which has repercussions for the democratization and understanding of science [28,29]. Furthermore, bearing in mind what has already been mentioned about most literacy being produced during formal education, education must play a relevant role in science teaching [30] to facilitate participation in decisions related to science and technology. One way this can be achieved is to equip the curriculum with a stronger presence of NoS to connect scientific knowledge with the social context and real life, so that pupils perceive such knowledge as relevant [31].

The term "citizen science" was coined by Irwin in 1995 to describe a type of collaboration between non-expert citizens and specialist scientists [32–34]. Although it still has little visibility in many developing countries, citizen science is assuming an increasingly important role as a tool for science and engagement [35]. Citizen participation can take on many forms: data collection, calculation, analysis, evaluation, the development of hypotheses, and methodology design or the diffusion of data [33,34]. Thus, even the most lay members of the public can contribute to the construction of science by doing such things, thereby making citizen science a democratizing element for science and scientific policy [36,37]. Citizen science can, therefore, make major contributions to informal scientific education, through which citizens are involved in real and significant scientific research [34,38–40]. As a result, citizens can improve their scientific knowledge [38,41]. Moreover, providing access to the generated data has the potential to empower people and change the ways in which citizens interact with each other and with their surroundings [42–44].

Some of the effects of citizen science on scientific literacy can be measured from the results of the participants in terms of [45]:

- Improvements in comprehension of scientific contents;
- Improvements in comprehension of scientific processes;
- Improvements in attitudes towards science;
- Improvements in abilities to engage in science;
- Increases in interest in science as a career.

Citizen science, furthermore, actively contributes to achieving the SDGs [7,46,47] as it covers a wide range of relevant areas (for example, water and air quality, marine debris, biodiversity, and health and gender issues), providing data for the achievement indicators of the set standards [48]; in this way, citizen participation is crucial for obtaining data and tracking the changes produced at a global level. Such data should be freely accessible so that citizens can make informed decisions [49]. The SDGs were approved by the United Nations in 2015, within the 2030 Agenda for Sustainable Development, to ensure continuity with the Millennium Development Goals and complete what those goals did not achieve. The SDGs are also focused on areas or pillars of critical importance to humanity and the planet [47]:

- People: end poverty and hunger and ensure that everyone can develop their potential with dignity and equality in a healthy environment;
- Planet: protect the planet from degradation through responsible consumption and production to guarantee support for the needs of future generations;
- Prosperity: ensure that all humans can enjoy a prosperous life with an economy in harmony with nature;
- Peace: strengthen a peaceful, just, and inclusive society;
- Partnership: mobilize the necessary means to implement the Agenda with the participation of all countries, stakeholders, and people.

Based on these aforementioned pillars, the 2030 Agenda includes 17 Sustainable Development Goals and 169 targets, divided into a total of 232 indicators that are used to monitor the progression in achieving the relevant goals. The 17 SDGs are [47]:

- Goal 1. end poverty in all its forms everywhere;
- Goal 2. end hunger, achieve food security and improved nutrition, and promote sustainable agriculture;
- Goal 3. ensure healthy lives and promote well-being for all ages;
- Goal 4. ensure inclusive and equitable quality education and promote lifelong learning opportunities for all;
- Goal 5. achieve gender equality and empower all women and girls;
- Goal 6. ensure the availability and sustainable management of water and sanitation for all;
- Goal 7. ensure access to affordable, reliable, sustainable, and modern energy for all;
- Goal 8. promote sustained, inclusive, and sustainable economic growth, full and productive employment, and decent work for all;
- Goal 9. build resilient infrastructure, promote inclusive and sustainable industrialization, and foster innovation;
- Goal 10. reduce inequality within and among countries;
- Goal 11. make cities and human settlements inclusive, safe, resilient, and sustainable;
- Goal 12. ensure sustainable consumption and production patterns;
- Goal 13. take urgent action to combat climate change and its impacts;
- Goal 14. conserve and sustainably use the oceans, seas, and marine resources for sustainable development;
- Goal 15. protect, restore, and promote the sustainable use of terrestrial ecosystems, sustainably manage forests, combat desertification, halt and reverse land degradation, and stop biodiversity loss;
- Goal 16. promote peaceful and inclusive societies for sustainable development, provide access to justice for all, and build effective, accountable, and inclusive institutions at all levels;
- Goal 17. strengthen the means of implementation and revitalize the global partnership for sustainable development.

The SDGs are interconnected and integrated, such that contributing to the development of one goal contributes to the development of the others. However, to relate the different SDGs to each other, citizens must acquire key sustainability competencies. These competences cannot be taught, and must be acquired by the learner through experience and reflection. Education for Sustainable Development (EDS) [50] aims to develop cross-cutting sustainability competencies among learners for the achievement of these goals, by transforming each learner's own behavior. Thus, education, which is a goal in itself, represents an essential strategy for achieving the SDGs [50,51].

The implementation of citizen science in the classroom, starting with its integration within the curriculum, should be enriched by seeking to achieve the objectives of scientific literacy and those of the SDGs. This should be done without losing the essence of a collaborative project stemming from a teacher-led process and complemented by the aspects that the project may lack, due to being designed as a citizen science project, rather than a project designed from the outset for teaching. However, some bodies, such as the Fundación Ibercivis, appreciate the need teachers have for educational materials that foster the implementation of such activities in the classroom. Such bodies offer guides for different projects that they carry out in the form of teaching units, such as the *OdourCollect Teaching Unit* [52] and the *Vigilantes of the Air Teaching Unit* [53], which act as an aid when working with citizen science at different educational levels, thereby respecting the approach.

AQUA is a citizen science project undertaken by Fundación Ibercivis and funded by the Spanish Foundation for Science and Technology (*Fundación Española para la Ciencia y Tecnología*, FECYT). The project's aim is to determine the quality of the water we drink and, from the data obtained, develop a map showing the points where the different parameters of household drinking water have been measured: chlorine concentration, degree of acidity or basicity, odor, and taste. To do this, the Fundación Ibercivis supply a kit comprising a test tube, an indicator tablet for chlorine concentration,

acidity indicator paper, and instructions for carrying out the test and collecting the data [54]. Later, each user can share their data and geolocate the sample site by means of a smartphone app or the project's webpage [55]. The results, once input on the webpage, can be freely used by any citizens who want to determine the water quality in their area, their region, or their country, thus empowering them to carry out their own research and obtain their own conclusions. In this way, a connection is provided between citizens and the scientific team, turning the citizens into participants, not only in the sampling and data collection process, but also in the large-scale viewing of the results, thus allowing citizen science to contribute, as indicted, to some of the SDG [46] achievement indicators. In this case, the water quality analyzed in the project is the main focus of Sustainable Development Goal 6 [7,46,47], with Goals 3, 4, 5, 11, and 15 being integrated into the scheme in an overarching way.

In particular, the students involved in the project use citizen science tools to monitor the levels of water quality, which is included in Target 6.3 of SDG 6, "By 2030, improve water quality by reducing pollution, eliminating dumping and minimizing release of hazardous chemicals and materials, halving the proportion of untreated wastewater and substantially increasing recycling and safe reuse globally", as well as Indicators 6.3.1, "Proportion of wastewater safely treated", and 6.3.2, "Proportion of bodies of water with good ambient water quality" [56]. This is an opportunity for students to produce useful data as "this kind of data collection could encourage citizens to adopt more water-efficient behaviours, by raising their awareness of the issues, giving them a sense of ownership and encouraging them to reflect on potential solutions" [57].

The subject of the AQUA project allows us to work on Goal 6, to understand the importance of water characteristics and the parameters that regulate its quality. Furthermore, it clearly connects with Goal 11, which focuses on the water supply systems of cities from a historical and social perspective, and reflects on the situation in many countries that do not have access to drinking water. On the other hand, Goal 15 relates the quality of water in the home and in rivers to the health of the environment. In a cross-cutting manner, the students work on Goal 3 through critical reflection on the importance of hygiene in health, and the important role that water plays. Goal 4 is developed through the design of inclusive cooperative activities in this citizen science project. Finally, Goal 5 is addressed by recognizing that, regardless of gender, all citizens must have the possibility of participating in the same scientific and cultural activities, a factor that is carefully considered in this project.

## 3. Objectives and Research Methodology

The objectives that we propose in this article are the following:
- Demonstrate the implementation of a citizen science project, AQUA, in the classroom, starting with its curricular integration and complemented with activities that deal with the development of the diverse dimensions of scientific literacy, particularly in aspects of NoS.
- Analyze the changes in attitude towards science and technology among the learners involved in undertaking the citizen science project.
- Analyze the improvement in scientific literacy of the pupils in the dimensions of scientific processes and scientific situations.
- Analyze the contributions of the citizen science school project towards achieving the SDGs.

### 3.1. Participants

The sample comprised 83 pupils in their second cycle of compulsory secondary education ($M_{age}$ = 14.70 years, $SD_{age}$ = 0.71), including 49 girls ($M_{age}$ = 14.67 years, $SD_{age}$ = 0.72) and 34 boys ($M_{age}$ = 14.74 years, $SD_{age}$ = 0.71), attending a city center school in Northern Spain. The socio-economic level of the students' parents was medium to high.

### 3.2. Instruments

Consent was obtained from the management at the school where the study took place. Likewise, all the participants and their families were informed of the objectives of the study, and their consent was obtained. Assigning the pupils to the sample was done using convenience sampling. To gather the data, a questionnaire was drawn up (Appendix A). This questionnaire was designed to achieve the objectives of the research by using the RecerCaixa questionnaire [58] and the validated forms used in the ROSE project [59]. The questionnaire contains one multiple choice question and 15 questions answered using a Likert type scale of value between 1 (totally disagree) and 4 (totally agree). The pupils filled out the questionnaire before and after participating in the citizen science project with a time interval of three weeks between the two questionnaires.

### 3.3. Data Analysis

A before-and-after quasi-experimental design without a control group was applied. Data analysis was done by using the SPSS v.24 statistics package, which calculated the mean, the standard deviation, and the Cohen's d coefficient.

### 3.4. Integration of the AQUA Citizen Science Project in the Classroom

There were two different stages in this study. The first comprised the actions required by the AQUA project, and the second included not only the activities designed to achieve the strengthening and acquisition of scientific competence—which benefited the pupils' scientific literacy—but also the presentation of the SDGs related to the project the students were undertaking. These stages, although overlapping, are described separately for the benefit of the reader in a schematic form below.

Stage I: implementation of the AQUA citizen science project. During the first stage, the pupils become familiar with some of the scientific processes. They use a kit featuring materials and scientific instruments, take measurements of quantitative and qualitative parameters, and record the relevant data, which will later be available online for all citizens to freely study and peruse.

1. Each pupil participating in the classroom is given a water analysis kit for their homes and an explanation of what the project consists of.
2. Each pupil must carry out measurements of the indicated water (chlorine concentration, acidity-basicity, odor and taste) to discover the quality of the water in their homes.
3. The participants geolocate the data by using the IberAqua [60] app or the project webpage [55], as shown in Figure 2.

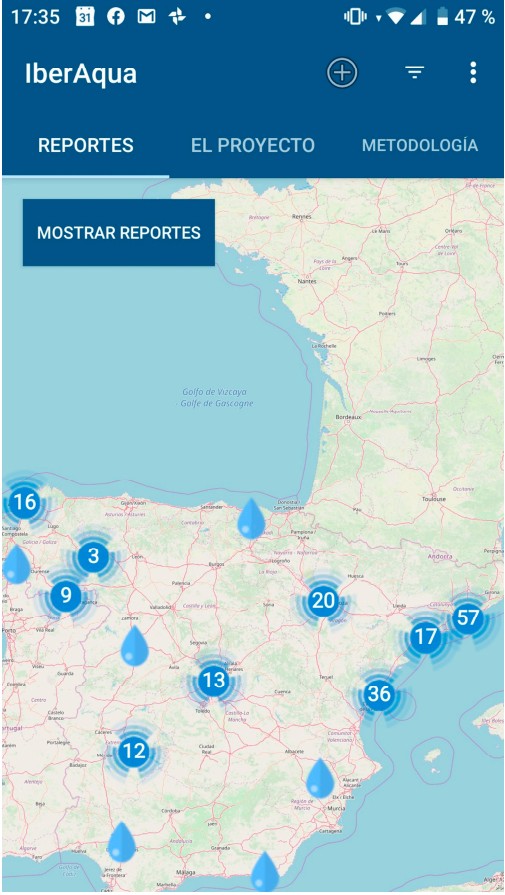

**Figure 2.** Pupils geolocate the origin of the parameter data obtained for the water using the IberAqua [60] Smartphone app.

Stage II: complementary activities. In this subsequent stage, the pupils were told to provide evidence of the experiments they had done by taking photos or videos. Keeping a record of the activities carried out is good scientific practice, but in this case, it is also a way to achieve the educational objective: preventing the pupil reproducing by rote the actions needed to perform the task (or even not doing it), and making sure the student uses the data obtained from their classmates' comments. Later, small teams were formed (small scientific communities) to analyze the data, to finally share and discuss their conclusions with the class. Once a consensus had been reached on how to interpret the data, the pupils, guided by the teacher, created educational materials (presentations and posters) to display in class and take their research to other classes and years at the school.

4. The pupils had to document the process by taking photos or making an explanatory video; the students then sent these multimedia documents to the teacher for assessment (Figure 3).

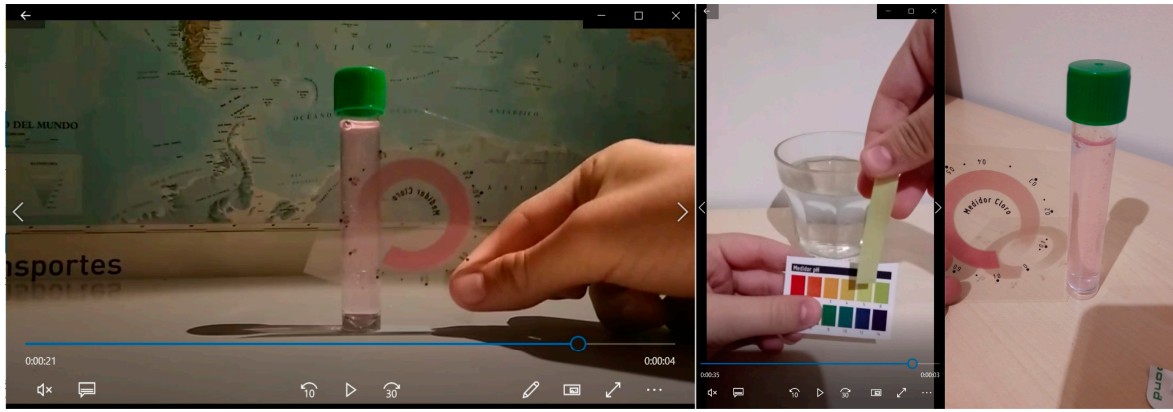

**Figure 3.** The pupils took videos and photos to document the activity. As seen in the images captured, the students carried out analysis of the water in their homes, thereby determining the acidity degree and chloride concentration.

5. Later, after mapping the city with the data provided by the participants, the pupils, in groups, had to analyze the average results of the water quality parameters for their city and compare them with the other regions in the country.

6. Finally, the teams had to create posters and promotional elements to show their project to their classmates and those in different school years (Figure 4).

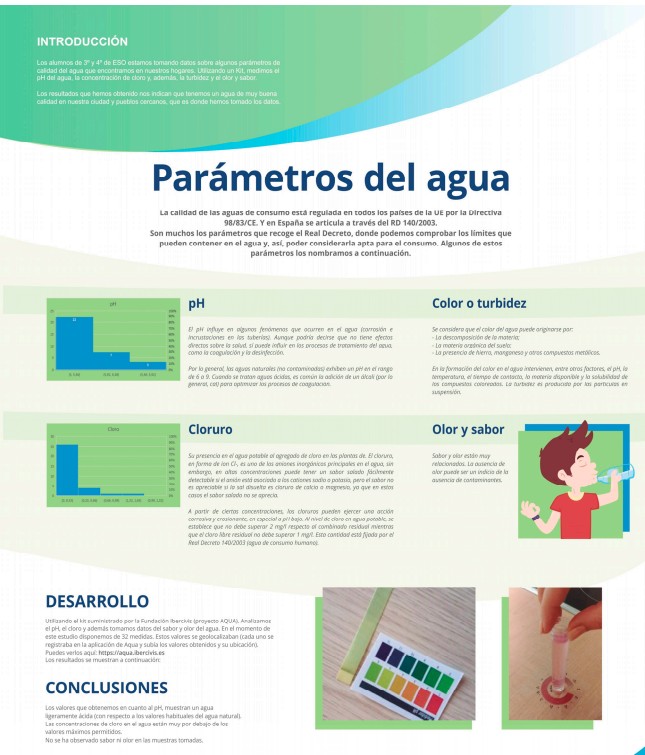

**Figure 4.** Pupils designed elements to display their results, such as an image poster, which they designed for their presentations and for use in science fairs.

The activity was undertaken over five sessions, which are briefly described below:

- Session 1. Pupils filled out the initial pre-test form. Their forthcoming activity was explained to them. The teacher provided them with the kit to perform the tasks at home. The students analyzed the water and recorded the results. At the same time, they documented the process with photos and videos. They uploaded the data onto the project's webpage by using the app or the webpage itself [60].

- Session 2. In teams of roughly four pupils, the students shared the results they obtained and discussed them. The spokesperson for each team explained the results they had gained to the rest of the pupils.

- Session 3. The pupils processed all the data from the region to obtain the average values and then discussed any discrepancies. They compared the water quality data with those from other regions. They also sought explanations and created display materials.

- Session 4. The pupils presented their results, explaining any possible discrepancies to the other teams. The discussions continued until a consensus was reached.

- Session 5. Exhibition and diffusion of the results to student from other school years at the center. The pupils then filled out the final post-test.

As the activity developed, the pupils were not limited to producing a technical study of the water parameters and discussing it, but instead were also able to study the water from different

angles: its physical and chemical properties, the water in the universe, the water on our planet, water's importance, the water we drink, sustainability, and water in the news, etc. Furthermore, the project was an integrating opportunity to provide a formal context to make the SDGs known to the students and allow the students to see how their work relates directly to Sustainable Development [7,46,47] Goal 6, with Goals 3, 4, 5, 11 and 15 included in an overarching way.

## 4. Results

To achieve the objective: *Demonstrate the implementation of a citizen science project, AQUA, in the classroom*, it is commonplace in all curriculums to have practical sessions focused towards realizing the pupils' learning outcomes and skill development [61–63]. By complementing the project as described, it was largely possible for the pupils to acquire the appropriate level of learning specified in the curricula, and to develop the corresponding scientific skills.

For the objective *Analyze the changes in the attitudes towards science and technology among the learners involved in undertaking the citizen science project*, a series of icons representing different "types" of population (Appendix A) were shown in the pre-test and post-test, indicating that the pupils should respond to the following question: *Which person or persons in the images do you think could be a scientist?* The answers to this question are shown in Figures 5 and 6, the latter of which shows them separated by the gender variable.

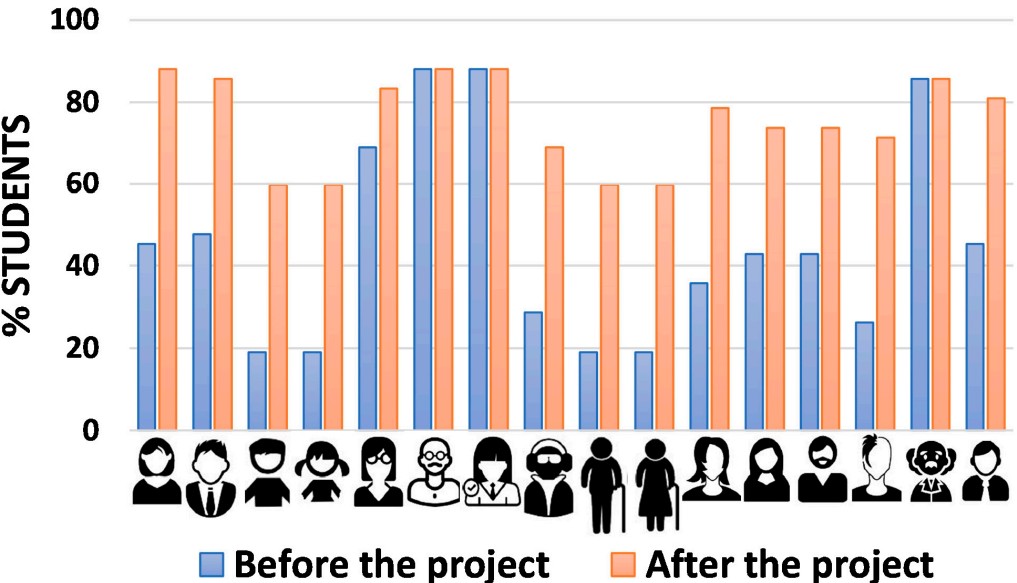

**Figure 5.** Responses to which icons were associated with scientists by the pupils: before and after. The graph shows the changes in student perception regarding the "scientific" stereotype. Character icons are courtesy of RecerCaixa [58].

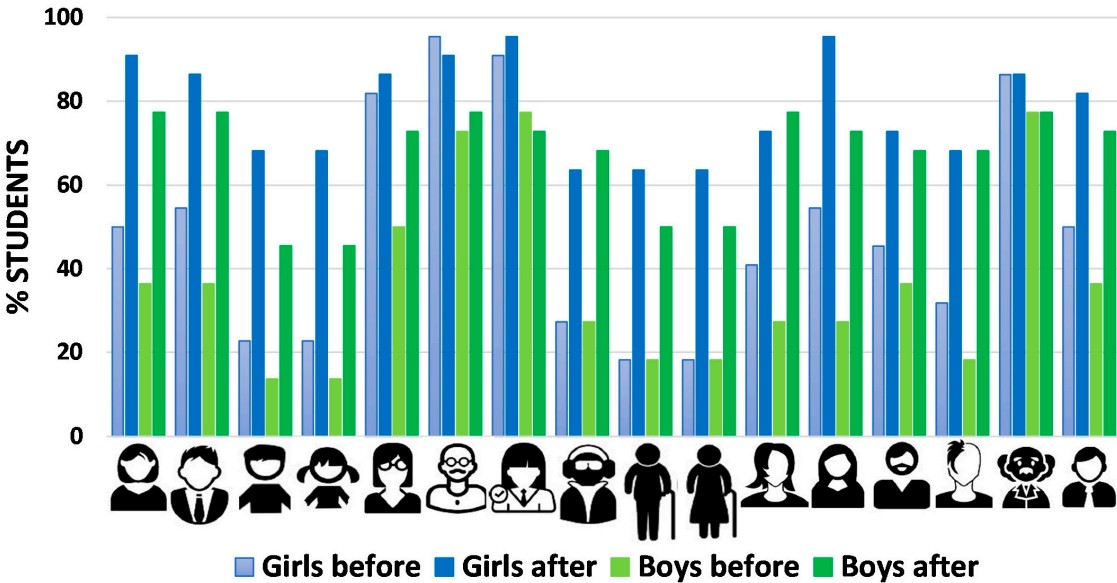

**Figure 6.** This graph shows which of the icons were associated with scientists by the pupils, before and after, broken down by gender. It can be seen, in general, that the stereotype about "who can be a scientist" is lower for girls, both before and after. Character icons courtesy of RecerCaixa [58].

Considering the results, a major change is appreciable in the shift in perception among pupils, regardless of the gender variable, regarding who can participate in science. However, the girls held a prior notion that *anyone can do science* (which is related to SDGs 4 and 5 [47]), and not just those represented by the icons showing traditional images of scientists. The pupils linked being a scientist to age, according to the representative icons. Nevertheless, as mentioned, a major change was noted.

The results of the questions (See Appendix A) applied to achieve this research objective are shown in Table 1.

**Table 1.** Data obtained for the "attitude towards science and technology" for all students and separated by gender, both before and after the educational intervention in the citizen science project.

| | Total | | | | | Girls | | | | | Boys | | | | |
|---|---|---|---|---|---|---|---|---|---|---|---|---|---|---|---|
| | Before | | After | | | Before | | After | | | Before | | After | | |
| Question | M | SD | M | SD | d | M | SD | M | SD | d | M | SD | M | SD | d |
| A1 | 2.77 | 0.75 | 3.52 | 0.70 | 1.03 | 2.67 | 0.77 | 3.43 | 0.82 | 0.96 | 2.91 | 0.71 | 3.65 | 0.49 | 1.21 |
| A2 | 2.46 | 0.95 | 3.07 | 0.85 | 0.68 | 2.27 | 0.93 | 2.98 | 0.88 | 0.78 | 2.74 | 0.93 | 3.21 | 0.81 | 0.54 |
| A3 | 2.54 | 1.12 | 3.00 | 0.95 | 0.44 | 2.45 | 1.06 | 2.94 | 0.99 | 0.48 | 2.68 | 1.20 | 3.09 | 0.90 | 0.39 |
| A4 | 2.24 | 0.91 | 3.37 | 0.64 | 1.44 | 2.16 | 0.85 | 3.43 | 0.61 | 1.72 | 2.35 | 0.98 | 3.29 | 0.68 | 1.11 |
| A5 | 2.16 | 0.96 | 2.94 | 0.93 | 0.83 | 2.16 | 0.94 | 2.94 | 0.88 | 0.86 | 2.15 | 0.99 | 2.94 | 1.01 | 0.79 |
| A6 | 2.07 | 0.99 | 3.18 | 0.83 | 1.22 | 2.00 | 1.00 | 3.16 | 0.83 | 1.26 | 2.18 | 0.97 | 3.21 | 0.84 | 1.14 |

Note: M = mean, SD = standard deviation, d = Cohen's d. A1 = the sciences at school are interesting; A2 = the sciences at school are easy to learn; A3 = I like the sciences more than other subjects; A4 = what I learn in science classes helps me in my everyday life; A5 = the sciences at school have made me more critical; A6 = school science has increased my curiosity towards things that we still cannot explain.

The results show a positive change in attitude towards science and technology, and—bearing in mind that an effect size (Cohen's d) of around 0.5 represents a moderate effect, and one greater than 0.8 represents a large effect [64]—this change in attitudes and perceptions toward school science is significant, because the size of the effect is large for most of the items. Participation in the citizen science project integrated into the curriculum improved the pupils' interest, curiosity, and appreciation for the sciences (moderate effect size) in such a way that they perceived the sciences as

easier to learn (moderate effect size) after the project, in addition to seeing them as something useful in everyday life.

Table 2 shows the responses obtained to the questions corresponding to the aim *Analyze the improvement in scientific literacy of the pupils in the dimensions of scientific processes and scientific situations* (See Appendix A).

**Table 2.** The results obtained for "opinions on science and technology" for the total student body and separated by gender, taken both before and after the educational intervention of citizen science.

| Question | Total | | | | | Girls | | | | | Boys | | | | |
|---|---|---|---|---|---|---|---|---|---|---|---|---|---|---|---|
| | Before | | After | | | Before | | After | | | Before | | After | | |
| | M | SD | M | SD | d | M | SD | M | SD | d | M | SD | M | SD | d |
| B1 | 2.90 | 0.76 | 3.51 | 0.61 | 0.89 | 2.86 | 0.68 | 3.45 | 0.61 | 0.91 | 2.97 | 0.87 | 3.59 | 0.61 | 0.83 |
| B2 | 2.89 | 0.77 | 3.55 | 0.52 | 1.01 | 2.90 | 0.68 | 3.49 | 0.54 | 0.96 | 2.88 | 0.88 | 3.65 | 0.49 | 1.08 |
| B3 | 2.49 | 0.93 | 3.36 | 0.71 | 1.05 | 2.41 | 0.93 | 3.35 | 0.66 | 1.17 | 2.62 | 0.92 | 3.38 | 0.78 | 0.89 |
| B4 | 2.07 | 0.99 | 3.18 | 0.83 | 1.22 | 2.63 | 0.83 | 3.39 | 0.67 | 1.01 | 2.79 | 0.64 | 3.32 | 0.59 | 0.86 |
| B5 | 2.57 | 0.97 | 3.41 | 0.75 | 0.97 | 2.41 | 1.00 | 3.37 | 0.83 | 1.05 | 2.79 | 0.88 | 3.47 | 0.61 | 0.90 |
| B6 | 2.71 | 0.77 | 3.64 | 0.51 | 1.42 | 2.55 | 0.65 | 3.67 | 0.47 | 1.98 | 2.94 | 0.89 | 3.59 | 0.56 | 0.87 |
| B7 | 2.45 | 0.87 | 3.42 | 0.77 | 1.18 | 2.33 | 0.83 | 3.35 | 0.78 | 1.27 | 2.62 | 0.92 | 3.53 | 0.75 | 1.08 |
| B8 | 2.53 | 0.89 | 3.45 | 0.63 | 1.19 | 2.47 | 0.84 | 3.37 | 0.64 | 1.21 | 2.62 | 0.95 | 3.56 | 0.61 | 1.18 |
| B9 | 2.40 | 0.83 | 3.54 | 0.65 | 1.53 | 2.31 | 0.80 | 3.55 | 0.68 | 1.67 | 2.53 | 0.86 | 3.53 | 0.61 | 1.34 |

Note: M = mean, SD = standard deviation, d = Cohen's d. B1 = science and technology are important for society; B2 = thanks to science and technology, there will be greater opportunities for future generations; B3 = science and technology make our lives healthier, easier, and more comfortable; B4 = the sciences at school have increased my respect for nature; B5 = the sciences at school have taught me to take better care of my health; B6 = new technologies will make work more interesting; B7. scientists follow the scientific method, which always leads them to improve their responses.

The results in this case also show a positive change (with a large effect size ($d > 0.8$)) in the pupils' scientific literacy in the aspects under analysis.

For the gender variable, the size of the effect is greater in the responses from girls than in those from boys, with the exception of the question *Thanks to science and technology, there will be greater opportunities for future generations (B2)*, although the difference is very small. The effect size values obtained in the responses to the question *New technologies will make work more interesting (B6)* are of note, for which the girls achieved a value of $d = 1.98$ compared to their male classmates with $d = 0.87$.

For the objective *Analyze the contribution the citizen science school project has made towards achieving the SDGs*, as mentioned, citizen science contributed to achieving the SDGs, not only because of the interest in the topic studied (the quality of water from Goal 6 [47]), but also because the information given to the pupils meant that they were able to develop an informed opinion, based on the data obtained in the project. Incorporating all this into the curriculum turned the youngest pupils into participants who, through the activities (sharing, debating, presenting, and diffusing), were able to form an opinion and, furthermore, act as a focal point for the dissemination of the results. This was shown by the pupils' responses to the item *The sciences at school have made me more critical (A5)* (Table 1). At the same time, as shown in Table 2, the variation in the scores obtained for the items *Science and technology make our lives healthier, easier, and more comfortable, The sciences at school have increased my respect for nature (B4)*, and *The sciences at school have taught me to take better care of my health (B5)* indicates a change in the pupils' perception of the environment and their own health (Goal 3 and Goal 15) [47]. Furthermore, all the students worked on an inclusive, safe, resilient, and sustainable project (Goal 4, 5, and 11) [47], in which everyone, in equal measure, acted as a scientist. This can be seen in the change in appreciation for the genders and ages of people who can be involved in scientific processes, as shown in Figures 5 and 6.

## 5. Discussion and Conclusions

Some areas of scientific literacy, particularly those regarding NoS, are not adequately implemented in the classroom. This is sometimes due to teachers not being (or feeling) qualified, not believing such areas to be relevant, or not knowing how to integrate them into the relevant science subjects [5,65] on the curriculum, even though comprehension of the NoS is a key skill that citizens must acquire as part of their scientific education [4,66]. This article described a way to integrate a citizen science project into the curriculum, so that relevant citizen science achievements are included within formal education. The way to do that, is connecting those achievements with students, particularly those aspects not always found in science teaching but important to foster a culture of science, such us understanding how science is built and its methods. As a complement to this citizen science project, other vital aspects of scientific literacy could also be studied in depth, such as the social character of science and the scientific community, and the construction of scientific knowledge on the basis of a consensus reached through debate and justification. Thus, the importance of the ideas of the scientific community in the design of effective policies for sustainable development [67,68] and the importance of diffusing those results is very clear. Some of the dimensions of scientific literacy [Figure 1] are only developed by undertaking additional activities that complement the citizen science project, such as those described in this article: procedural dimension (PD) 1 (self-learning science), including reflections about the scientific process and the analysis and processing of the data in teams, as well as the feedback obtained from classmates and the teacher; PD5 (encode scientific communications), with the documents generated by the pupils to subsequently communicate the results (with presentations and posters); PD6 (think scientifically), getting and processing data and sharing them with work teams, as scientific communities do; PD7 (reason and argue), analyzing the data to debate with classmates and justify your own beliefs through experimental facts; PD8 (judge the validity of claims), using debates held by the students and assessing how they argue their points; PD9 (make decisions), choosing the best options to achieve the best results and finding the most suitable ways to report them.

Scientific literacy will presumably make it possible to develop critical thinking among pupils, and promote citizens who are freer, and have greater ownership of their destiny, which is what scientific education aims for [31]. Pupils must learn about science in the same way that science itself works [23]. Activities for investigating and experimenting in the classroom are suitable, but they do not cover all aspects of NoS or scientific literacy [3]. In the citizen science project, the pupils work with real data, and make a contribution to a real scientific project set in a context that is much broader that the school setting and transcends the limits of the school. This is a real project in a real world; the pupils are not working in a simulation, but instead form part of a global project that constructs science in the same way that real science is created, by cooperating and working in teams, justifying and discussing, and communicating and evaluating their ideas [1]. Other types of projects or activities developed in the classroom elucidate the work of scientists, but do not let the pupil experience the process of constructing science or contribute at the same time to advances in the SDGs as we achieved with the classroom integration described in this proposal, which contributed to achieving Goals 3, 4, 5, 6, 11, and 15 [47] as follows: Goal 3, through the students' critical reflection on the importance of water and personal hygiene habits for health; Goal 4, participating in cooperative inclusive activities and helping to create a more sustainable world through participation in the project of citizen science; Goal 5, recognizing that, independent of gender, all citizens have to have the ability to participate in the same activities, both scientific and cultural; Goal 6, comprising the importance that water plays in our lives and activities by understanding the characteristics of the water and the parameters of regulating its quality; Goal 11, understanding the systems of the water supply in the city from an historical and social perspective, and how water remains a problem in many countries; Goal 15, relating water quality at home and in rivers to the health of the environment.

Other experiments for implementing citizen science projects in the classroom, such as the RecerCaixa [58] pilot experiment mentioned previously, require more complex structures and a number of teachers and science staff involved, which makes implementing them in every school unfeasible or at least complicated. In this experiment [58], the differences between the pre-test and

the post-test are not obvious in the responses for which icons the pupils associate with a scientist. Future research should study this aspect, because one of the most relevant elements of citizen science is that the people participating are involved in real scientific research [34,38–40], which means that they should see themselves as scientists who construct science. It is possible that, in this case, the pupils did not make a strong association or connection. Importantly, research methodologies that involve self-directed student learning are still scarce in many of our classrooms. In [69], a project was developed with university-level science students, by compelling them to take the first years of their education in non-scientific specialties; a positive change was also obtained in their perceptions of science and technology, as well as an increase in their interest toward participating in science projects.

This article showed changes in the perception of science and technology among the pupils involved, which agrees with other studies on the impact of citizen science, both in formal [58,69] and non-formal education [34,38,40,45], as well as in pilot trials for the co-creation of citizen science projects with secondary pupils [40,67]. Likewise, activities like these seem to improve the scientific literacy of the pupils involved in the aspects mentioned in this article. However, it would be complex to integrate these proposals for the co-creation of citizen science projects into all school centers, as it is impossible for all schools to work together with scientists [4]. Furthermore, offering support to citizen science projects at a local level, by integrating them into the formal education curriculum, will guarantee that such projects will contribute to reports on the SDGs and, at the same time, facilitate social innovation, through which citizens can help monitor and implement the SDGs [46,70], thus inspiring the citizens (pupils) to become agents of change and develop new sustainable ways of living [71]. At the same time, the SDGs are interconnected [72], which means that working to develop some of the goals favors the development of the others. However, one of the great challenges facing the scientific community is to demystify the process of science, and translate the process and its results for consumption by non-scientific citizens [73]. As this article has shown, the implementation of citizen science in school can facilitate this process and engender awareness and commitment among the pupils. A significant challenge for schools is not only to achieve scientific literacy among pupils who will pursue science degrees, but also among those that are not going to pursue scientific subjects, as a pupil's experiences with science, along with the related prejudices and suppositions, are limited to those formed during his or her school years [3].

On the other hand, education is essential, since education can develop cross-cutting key competencies for sustainability that are relevant to the achievement of all the SDGs [50,51]. Therefore, it is necessary to modify our educational approaches. The integration of citizen science projects provides an important framework for incorporating these goals at school.

**Author Contributions:** Research, M.A.Q.-D., E.L.-I., M.D.-O. and M.C.S.-M.; methodology, M.A.Q.-D., E.L.-I., M.D.-O. and M.C.S.-M.; supervision, J.B.V-D.; writing—original draft, M.A.Q.-D.; writing—review and editing, M.A.Q-D., E.L.-I., M.D.-O., M.C.S.-M. and J.B.V.-D; funding acquisition, M.A.Q.-D. and M.C.S.-M.. All authors have read and agreed to the published version of the manuscript.

**Funding:** This research was funded by the *Consejería de Educación de la Junta de Castilla y León* through the *Dirección General de Innovación y Equidad Educativa*, grant number EDUCYL2018_04 under the project «*Igualdad de género en las materias científico-tecnológicas: análisis comparativo estudiantes STEM*».

**Conflicts of Interest:** The authors declare no conflict of interest.

## Appendix A

### AQUA CITIZEN SCIENCE PROJECT QUESTIONNAIRE

**Put an X in the correct box.**

| Age | | | Sex: | Female | Male | | Year: | |
|-----|-----|-----|------|--------|------|-----|-------|-----|

1. Imagine that these images or icons represent people:

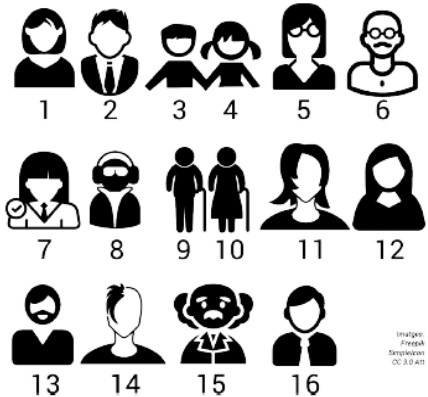

Indicate which of the people in the images you consider could be a scientist

| 1 | 2 | 3 | 4 | 5 | 6 |
|---|---|---|---|---|---|
| 7 | 8 | 9 | 10 | 11 | 12 |
| 13 | 14 | 15 | 16 | | |

2. Indicate how much you agree with the following statements, from 1: I disagree very much, to 4: I agree a lot.

| **Attitude towards science and technology** | **1  2  3  4** |
|---|---|
| A1. The sciences at school are interesting. | |
| A2. The sciences at school are easy to learn. | |
| A3. I like the sciences more than other subjects. | |
| A4. What I learn in science classes helps me in my everyday life. | |
| A5. The sciences at school have made me more critical. | |
| A6. School science has increased my curiosity towards things that we still cannot explain. | |

| **Opinion on science and technology** | **1  2  3  4** |
|---|---|
| B1. Science and technology are important for society. | |
| B2. Thanks to science and technology, there will be greater opportunities for future generations. | |
| B3. Science and technology make our lives healthier, easier, and more comfortable. | |
| B4. The sciences at school have increased my respect for nature. | |
| B5. The sciences at school have taught me to take better care of my health. | |
| B6. New technologies will make work more interesting. | |
| B7. Scientists follow the scientific method, which always leads them to improve their responses. | |
| B8. We should always trust in what scientists have to say. | |
| B9. School science has shown me the importance of science for our way of life. | |

**Character icons courtesy of RecerCaixa.**

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
