# Peer review of "Citizen Science for Scientific Literacy and the Attainment of Sustainable Development Goals in Formal Education"

_sustainability, doi:10.3390/su12104283_

Round 1

Reviewer 1 Report

The article "Citizen science for scientific literacy and the attainment of SDGs in formal education" addresses a topic of interest today, when more and more researchers are convinced that education can make an essential contribution to global sustainable development. My perception is that beyond the research conducted the article is more a document of awareness of the importance of education in global sustainable development. The research objectives are generous, from demonstrations and analyzes in favor of the idea, until the change of attitudes is highlighted.

The structure chosen for dissemination follows the editorial recommendations. This includes the main parts: introduction, theoretical framework, objectives and research methodology, results, discussions and conclusions. The authors also put in two specific parts: the description of a project and the integration of that project in the class. I appreciate the innovation, but an integration of the additional parts in the main ones could contribute to facilitating the understanding of the scientific approach.

The six figures and two tables come to complete the understanding of the approach. An express account of the sources, positioned maybe under the title of the figures, respectively the tables, would be welcome.

To be appreciated is Annex A, which presents the questionnaire used in research and which can be a good example for those who want to replicate research in other environments. It is worth noting the explanatory details regarding the contribution of each of the five authors.

The details regarding the financing of this research come to complete the perception regarding the article's contribution to the main flow of scientific papers in the field.

Author Response

Thank you very much.

Reviewer 2 Report

  • A really intersting reserach and project. Work hand by hand with students at young ages is very useful for science literacy. Good idea!
  • Authors should include more information about the 17 Sustainable Development Goals (mainly background and why are so important and well-known nowadays).
  • Authors should improve the way they use to justify their research to the achivement of th SDGs (wich ones are involved with this work and how they make a significant contribution). It is important to higlight that they are doing a good contribution.
  • Well use of graphic resourses (if it is possible, I recommend to improve the quality of the figures).

Author Response

Thank you very much.

Reviewer 3 Report

The manuscript by Queiruga-Dios et al. demonstrates how a citizen science project can be integrated in the education and how this affects their perceptions, which in turn serves to fulfil Sustainable Development Goals proposed by the UN. The work is well conducted and addresses important issues of natural science, integration of citizens and potential future educational approaches.

Specific, minor comments:

Title: please use the full-length term for SDG as the abbreviation is not familiar to all readers

Lane 220: … once input….

Lane 268: was consent only requested or also obtained?

Lane 267: What was the evaluation time of the research project (time between before and after questionnaire)

Lane 287: please delete a) – it is fine with Stage I (the same applies for lane 300)

Lane 311: delete the first word (Furthermore)

Lane 321: bracket missing before Figure 4

Lane 382: I have the impression the correct value for large effects is 0.8 and not 0.08

Lane 386: it would be good to have a brief text describing the effects seen (not only the questions A1-A6). The same for Table 2. Or put the questions directly next to the table. It is not very convenient to consult the appendix.

Lane 413: bracket missing after Goal 15

Lane 498: citizen science in/at school?

Reviewer 4 Report

In this manuscript the authors present results of a curricular integration of citizen science in formal teaching, showing attitude changes of pupils towards science and technology and improvement in scientific literacy in terms of scientific processes and situations. I am coming late to this review process, but I see that significant edits based on previous recommendations have already been addressed, making this manuscript a joy to read!

1) Major points:

No major points.

In my opinion this is a very thoughtful, written, and properly references manuscript. 

2) Minor points:

Looking through the manuscript I see that all figures appear very grainy (and not be in the highest resolution possible) in the provided preview. I would be surprised if these figures would be published in the present quality. Perhaps this was only due to my preview. I am wondering if MDPI can help with this?

For Figure 1: Although I understand that this figure has been adapted from Kemp (Ref 6), it appears very grainy and I find it hard to read. Is it possible to provide the same information in an own, updated, high-resolution figure?

Line 121: to all citizens. I’d suggest to replace “to all citizens.”  With “all members of the society”. Not all citizen scientists are citizens. Citizen Science specifically caters to and draws in marginalized groups, and that includes many individuals with some kind of immigration status, therefore are not citizen per se. (This brings up a debate if community science would be a better term to describe individuals participating in science proper, but not classically trained in science proper. I personally see no issues using Citizen Science here for his manuscript, but would like to make the authors aware of this debate and distinction for future publications.)

Lines 138-140: It has been argued that one reason for the low visibility of citizen science research are the difficulties associated with publishing data generated by non-stakeholders. If appropriate, I’d suggest to include the following references in line 140 https://doi.org/10.22323/2.17030101 and https://doi.org/10.1016/j.biocon.2016.05.014.

Lines 315 and 316- delete "their", as students measure samples and not themselves.

As it can be seen in the images captured, the students carry out the analysis of the water in their homes, determining degree of acidity and concentration of chloride.

I see some minor grammatical issues throughout the manuscript, but nothing that cannot be edited in final typesetting step. 

Text spacing issues in lines 411 and 412
